# Milankovitch, the father of paleoclimate modeling

Andre Berger

Georges Lemaître Center for Earth and Climate Research, Earth and Life Institute, Université catholique de Louvain, Louvain-la-Neuve, 1348, Belgium

*Correspondence to*: Andre Berger (andre.berger@uclouvain.be)

**Abstract.** The history of the long-term variations of the astronomical elements used in paleoclimate research shows that, contrary to what might be thought, Milutin Milankovitch is not the father of the astronomical theory but he is definitely of paleoclimate modeling. He has not calculated himself these long-term variations but used them extensively for calculating the "secular march" of incoming solar
radiation. He has advanced our understanding of Quaternary climate variations by two important and original contributions fully described in his Canon of Insolation. These are the definition and use of caloric seasons and the concept of the "mathematical climate. How his mathematical model allowed him to give the caloric summer and winter insolation a climatological meaning is illustrated.

## 1 Introduction

Paleoclimatology is primarily a reconstruction of past climatic variations on the basis of proxy records. It aims also to explain these variations from principles of climatic behavior using climate models. Milankovitch has contributed significantly to this second objective by using the astronomical parameters to compute the long-term variations of his caloric insolation which he used in a climate model (although very simple) to reconstruct the past climates. This paper intends to underline the fundamental and original
contributions of Milankovitch to the understanding of the long-term climatic variations over the last one million years.

The two remarkable books of Milutin Milankovitch, his 1920 Théorie Mathématique written in French and - his 1941 Kanon der Erdbestrahlung written in German and translated in English in 1969, have
largely contributed to his reputation. The celebration of the 100th anniversary of his 1920 French book is a good opportunity to stress what were his main contributions and to "rendre à César ce qui appartient à César (give back to Caesar what belongs to Caesar)". There is indeed a tendency to over-inflate one's work for reasons that have nothing to do with the scientist, but for reasons that have to do with corporative or national politics of history of science. It is important to stress here that Milankovitch was always very
careful through all his publications referring properly to the publications of others when he was using their results.

## 2. Long-term variations of the astronomical parameters

Contrary to what might be thought, Milankovitch did not calculate the long-term variations of the astronomical elements. He used them extensively for calculating the "secular march" of the incoming solar radiation.

The first to calculate these astronomical elements was the French astronomer Joseph-Louis Lagrange (1736-1813). He made the calculation for the six great planets (1781-1782). At that time, Uranus had not been discovered and the mass of the planets could only be roughly estimated. Aware of this uncertainty of the masses, Lagrange investigated its possible influence on his calculations, a formulation that was going to be used more than one century later by Vojislav Michkovitch (1892-1976), a colleague whom Milankovitch solicited his collaboration.

During the early 19th century, Pierre-Simon Laplace (1749-1827) wrote his 5-volume Celestial Mechanics between 1799 and 1825. Philippe Gustave le Doulcet, Comte de Pontécoulant (1795-1874) carried out the computation of the long-term variations of the elements of the great planets but with a few decimals only.

It is during the second part of the nineteenth century that Urbain Le Verrier (1811-1877) introduced a new theory of the planetary motion (1855) and the calculation of the secular perturbations (1856). He published the numerical values of eccentricity (with a precision of $10^{-4}$), longitude of the perihelion (in arc minutes), inclination (in arc seconds) and longitude of the node over 100,000 years before and after 1800 A.D. each tenth millennium. His calculations were carried out before he discovered the planet Neptune. As this planet could not therefore be included in the Le Verrier calculations, John Nelson Stockwell (1832-1920) computed the secular perturbations by considering all the eight planets known at his time (1873): Mercury, Venus, the Earth, Mars, Jupiter, Saturn, Uranus and Neptune. Stockwell, like Le Verrier, had the possibility of correcting their computations by using better values of masses. This work completed the calculations of the secular perturbations of the great planets, but an error was discovered by Harzer (1895) twenty-two years later. Finally, it is the German Ludwig Pilgrim (1879-1935) (known mostly as a pioneer in colorimetry) who computed, at a time when Milankovitch was completing his doctoral degree thesis in 1904, the astronomical elements required for the computation of insolation. Pilgrim (1904) extended the numerical computations of eccentricity, obliquity and longitude of the moving perihelion, using the Stockwell integrals for every fifth millennium over 1,010,000 before and 40,000 years after 1850 A.D. (part of these values are in Milankovitch, 1920 p 223-225, in Köppen and Wegener, 1924 p 254-255 and a more complete list in Milankovitch, 1941 Table VIII p 254-258) and also for dates where the longitude of the perihelion was either 90° or 270° (Northern Hemisphere summer at perihelion or at aphelion). In Milankovitch's own words (Milankovitch, 1920 p 222; 1941 p 372), Pilgrim was the first to compute adequately the elements affecting the long-term variations of insolation, so that Milankovitch could use them for his research. Pilgrim tried also to treat the Ice Age mathematically, but according to Milankovitch, this treatment of the climatological part by Pilgrim was a failure.

Milankovitch clearly indicates that for the calculation of his incoming solar radiation (insolation in short), he used first Stockwell-Pilgrim's values of the eccentricity, obliquity and precession for the last 1,000,000 years before 1850 A.D... The insolation values for 55°, 60° and 65°N were published in Köppen-Wegener (1924, p 214) before the calculation was extended to other geographical latitudes and the new values published in his Mathematische Klimalehre in 1930 (Milankovitch, 1941 p 253).

Because of some errors in Stockwell (already detected by Harzer) and because he wanted to use the astronomical parameters based on the most reliable values of the planetary masses, Milankovitch decided to use the Le Verrier calculation including his corrections for the masses. For completing this work, he asked the collaboration of his colleague, Prof V. Michkovitch (1892-1976). Michkovitch made the necessary corrections of the masses following Le Verrier procedure and computed the long-term variations of eccentricity $e$, obliquity $\varepsilon$, and climatic precession $e\sin\Pi_\gamma$ , ( $\Pi_\gamma$ the longitude of the perihelion) for the past 600,000 years before 1800 A.D. with the following initial values:

$$e_0=0.0168 \quad \Pi_{\gamma 0} = 99°30' \quad \varepsilon_0=23°27'55''$$

All these values were published in Mathematische Klimalehre, 1930 and in Astronomische Mittel in 1938 and reproduced in Table IX of his Canon (Milankovitch, 1941 p 260-262). Milankovitch carefully noticed that the Hungarian scientist von Bacsàk (1870-190) draw his attention to two calculation errors, both related to $\Delta(e\sin\Pi_\gamma)$, one at 500 kyr BP and the other at 465 kyr BP, errors that Milankovitch took care to eliminate from his tables.

This use of two different astronomical solutions explains why we find in the early work of Milankovitch the astronomical values of Stockwell-Pilgrim and in his later work the values of Le Verrier-Michkovitch. The comparison of the insolation values that Milankovitch calculated from these solutions show a good agreement. Milankovitch concluded that a further improvement of the planetary masses using the formulation by Le Verrier would not change "the essential features of the secular course of insolation as I have calculated".

Milankovitch was however well aware that the solution by Le Verrier could not be extended over millions of years because of the limited accuracy of the perturbation calculation that was based upon classical mechanics, missing the Einstein relativistic displacement of the perihelion of the planets.

## 3. Periods of astronomical parameters

Milankovitch calculated in great detail the incoming solar radiation on the Earth, but seemed to have not been much interested in the astronomical periodicities themselves. From his table (based on Stockwell-Pilgrim), he simply deduced the average period of the oscillations of eccentricity as being 92 kyr, varying between 77 and 103 kyr. For precession, he found an average period of 21 kyr, varying between 16.3 kyr and 25.8 kyr. For the longitude of the moving perihelion, he explains that its irregularities are due to the

longitude of the fixed perihelion, but whether the perihelion has a mean motion remains an open question. For obliquity, he noticed that it "oscillates between extremely narrow limits" with a relatively stable period of 40 kyr varying between 38 and 45 kyr (Milankovitch, 1941, p 264, 269, 270). It is actually the French mathematician Joseph Alphonse Adhémar (1797-1862) who was the first in 1842 to deduce the value of 21,000 years for precession by combining the astronomical precession calculated from the value
of 50.1" per year of the French astronomer Jean-Baptiste Joseph Delambre (1749-1822) and the rotation of the terrestrial orbit calculated from the value of 11.83" per year of the French mathematician Louis Benjamin Francoeur (1773-1849).

The harmonics of precession, in particular those with a period of 19,000 and of 23,000 years – also found
by Hays et al. (1976) in their geological data -, and the 400,000-year of eccentricity were discovered by Berger (1973, 1978a) who calculated all the periods in the expansion of the long-term variations of the astronomical variables used in the calculation of insolation.

It is worth noting that Milankovitch was very much interested in obliquity probably because of the strong
obliquity signal in his caloric half-year insolation (see Figure 1). Milankovitch mentioned many times that the authors who were mainly stressing precession, were not sufficiently or not properly taking into account obliquity. Tables XII and XIII of his Canon give the change of the radiation of his Table XI for an increase of the obliquity by one degree (respectively in canonic units and in percent). These three tables were first published in his "The Problem of the Astronomical theories of the Ice Ages" in 1914 showing
in details for the first time the influence of obliquity upon insolation. To put it briefly, an increase of obliquity reduces the latitudinal contrast between the equator and the poles mainly in the annual irradiation and increases the seasonal one (with an augmentation of the summer irradiation and a reduction of the winter one), with a similar effect in both hemispheres.

In fact, it is not widely known that similar insolation computations were actually already done in the 19th century by the Englishman Sir John Frederick William Herschel (1792-1871) (published in 1832 for the total irradiation), by the American scientist Levi Witter Meech (1821-1912) (published in 1856 for the daily and seasonal irradiation at any latitude based on elliptical integrals), by the German mathematician Ludwig Christian Wiener (1826-1896) (published in 1877, he did the same as Meech with in addition
the total irradiation over different parts of the Earth) and by the Irish intellectual Joseph John Murphy (1827-1894). It is actually Murphy who was the first in 1869 to put forward the idea that a long, cool summer and a short, mild winter are the most favorable conditions for glaciations (a hypothesis totally opposite to Croll, 1864). This idea was taken up by the Austrian climatologist Rudolf Spitaler (1859-1946) half a century later (in 1921). It does therefore follow that it is not Milankovitch who originated
this principle as some authors have claimed and still claim calling it the "Milankovitch model". Milankovitch has actually popularized and spread the idea under the advice of Köppen (1941) who claimed : "the diminution of heat during the summer half-year is the decisive factor in glaciation" and also following the comments made earlier by Penck and Brückner (1909) and Brückner et al. 1925: "From the climatological point of view, glaciers are not favoured by severe winter…but by a mild winter and a
cool summer". It must also be noticed that in Köppen-Wegener (1924 p184 English edition) we find:

*"…two cause responsible for the growth of a glacier are huge amount of snow and low temperature, especially in summer…".*

## 4. Caloric Seasons

When dealing with the astronomical seasons, the long-term variations of both their total irradiation and their length must be taken into account. To accommodate this duality, Milankovitch introduced the caloric seasons. This concept of caloric seasons is one of the two most important and original contributions of Milankovitch. These divide the year into two equally long seasons, one of which – the caloric summer - comprises all days during which the irradiation at the given latitude is stronger than on any day of the other half-year – the caloric winter. Because the semi-major axis of the Earth's orbit, the sidereal period of revolution of the Earth around the Sun and, to an excellent approximation, the tropical year do not change with time, the length of these caloric seasons are exactly 182.6211 mean solar days when the tropical year is used. This, however, does not solve the problem completely because of the start and end of these half-year seasons change with time and because the double maximum and minimum characterizing the insolation in the intertropical regions.

Milankovitch noticed that he discovered these caloric seasons after his 1920 book on "Théorie mathématique" was published and he used them for the first time in Köppen-Wegener (1924, p 194 English edition) and in his contribution (Mathematische Klimalehre) to the 1930 Köppen-Wegener Handbook. Actually, Köppen and Wegener invited Milankovitch to contribute a text (p 193 ff in Köppen-Wegener English edition 1924) in which Milankovitch referred to his 1923 paper (p 194).

In his Canon (1941), Milankovitch devoted twenty pages to the "quantities of heat received by a latitude during a caloric summer and winter half-year". From the formulas that he developed, it is clear that, during their local season, the impact of the variations of obliquity is the same in both hemispheres and maximum in the high latitudes, whereas the impact of climatic precession is opposite in the two hemispheres and maximum at the low latitudes.

In Chap XX of his Canon, Milankovitch gave the numerical values of the caloric Northern and Southern hemisphere summer half-years for 1800 (Table XXIII) and over the last 600,000 years (Table XXV) in canonic units (the canonic units introduced by Milankovitch are the units obtained if the solar constant is the unit of solar radiation and if the unit of time is 100,000 instead of seconds). Since no hypotheses were introduced for these calculations, Milankovitch, who was convinced of the perennity of his work, decided to call his results "Kanon der Erdbestrahlung" (Canon of Insolation).

With Köppen's approval, Milankovitch preferred not to continue reproducing the numerical values of insolation themselves, but rather to transform them in fictitious latitudes, called the 65°N equivalent latitudes; these values and figures were first published in Köppen-Wegener (1924, English edition p 240-241) and in his Mathematische Klimalehre but the definition was already given in Milankovitch (1920, p 73). These latitudes are actually the present-day latitudes which received during the Northern Hemisphere

caloric summer half-year the same irradiation as 65°N in the past. A fictitious motion of these latitudes to the south corresponds therefore to an increase of the summer irradiation and to the north to a decrease of insolation leading possibly to a glaciation.

## 5. The mathematical climate

It must be stressed that the main contributions of Milankovitch were not only based on his insolation and radiation curves, but also on his mathematical computation of the thermal effects of the secular march of insolation, his so-called mathematical climate. The direct effect calculated if insolation only was varying was published in his 1930 Mathematische Klimalehre. From the Stefan-Boltzmann law and a grey body model with the reflective power of the surface and the absorption coefficient in infrared kept invariable, Milankovitch calculated the long-term variations of the mean temperatures of the caloric summer and winter half-years ($\Delta T = \Delta Q / 150$, $\Delta Q$ in canonic units).

If the ice cover and other feedbacks are taken into account, the indirect effects could be estimated. This was published in 1938 Astronomische Mittel where Milankovitch calculated first the altitude of the snow line (called snow limit by Köppen and Milankovitch, 1941 p 437), $H_i$, as a function of the caloric summer insolation. This calculation was based on the correlation between these variables according to Köppen snow limit data for different latitudes. According to his relationship, any variation of the summer irradiation by 1 canonic unit produces a shift of the snow limit altitude by 1m ($\Delta H_i = 1.09 \Delta Q_S$).

These relationships allowed Milankovitch to give the caloric summer and winter insolation a climatological meaning. This shows how much he was concerned by climate and its variations. As these relationships are simple and straightforward, Milankovitch did not published any additional tables and referred only to his tables providing the long-term variations of the caloric summer and winter half-year insolation. From his Table XXV, we can see that the deficit in summer radiation can reach 573 canonic units at 75°N, 22100 years ago, which according to his formula means a drop of the altitude of the snow line by more than 500m. Following the Köppen table of the altitude of the snow limit for different latitudes (Milankovitch, 1941, Fig 53 p 435 and Table XXVII p 437), the polar cap can then have extended from 75°N up to 65°N, it means covering an area 2.75 times greater. This kind of deficit can also be reached in the tropical latitudes with an accompanying lowering of the snow limit altitude which, as noted by Milankovitch (1941, p 471) "refutes the opinion expressed by some geologists that insolation cannot explain such displacement". Milankovitch also pointed out that "owing to such variations of the summer irradiation, the mean summer temperature dropped from time to time by more than 5° in the high and temperate latitudes of both hemispheres and even in the tropical latitudes".

## 6. Irradiation over the polar caps

Most important is that such an increase of the size of the polar snow cover changes the reflective power of the Earth. This is why to complete hisCanon, Milankovitch decided to compute the long-term variations of the mean summer and winter insolation per unit surface area of the Northern and Southern polar snow-

caps over the last 600,000 years (Table XXVIII). The extent of these polar caps was deduced from the treatise by Wundt (1933). In this treatise, the Northern cap extends presently to 75°N and reached 55°N at the maximum of the Ice Age. From these values, Milankovitch could compute the long-term variations of the insolation over such polar caps delimited by the parallel 55° assuming that the extension of the snow cap was always proportional to the corresponding deficit in summer radiation. If the albedo is kept constant, it can be seen that the minimum summer radiation over the Northern cap reaching 55°N occurred 230,000 years ago, with a radiation deficit $\Delta_1 Q_S = 363\Delta\varepsilon - 7900\Delta\left(e\sin\Pi_\gamma\right),$, compared to the present, amounting to 660 canonic units ($\varepsilon$ is the obliquity, $e$ the eccentricity and $\Pi_\gamma$ the longitude of the perihelion). As this deficit caused a southward extension of the cap of 20°, Milankovitch concluded that a change by one canonic unit corresponds to a meridional change in the extent of the Northern polar cap by about 1´82 or 3.37 km. This means also that the Northern snow cap totally disappears for an increase of the summer insolation by 495 canonic units. This occurred quite a few times over the last 600,000 years, as for example 10,000 and 127,000 years ago.

Taking into account the reflective power of snow at the Earth' surface, for the cap reaching 55°N, $\Delta_2 Q_S = 0.802\left(13520 + \Delta_1 Q_S\right)\left[\sin\left(75° + 1´82\Delta_1 Q_S\right) - \sin 75°\right]$ shows that large negative amplitudes occurred several times over the last 600,000 years. At 230 kyr BP, the total deficit $\Delta Q_S = \Delta_1 Q_S + \Delta_2 Q_S$ amounts now 2180 canonic units. This is far more than the 660-deficit calculated if the reflective power of snow is not taken into account. It corresponds to a displacement downward of the snow limit of 2180 m which is about the present altitude of the snow limit at 55°N. This implies that the polar cap must have reached this latitude at that time, what according to Milankovitch was actually observed in the geological reconstruction. It is also interesting to notice with Milankovitch that the deficit of the annual radiation at 230 kyr BP amounted 1920 canonic units which means a decrease in the annual temperature by 6.4 °C ( $\Delta T_T = \dfrac{\Delta Q_S + \Delta Q_W}{300}$ ), contradicting those who claims that the long-term variations of insolation can not cause any important drop in the annual temperature.

Using the data for the cap reaching 45°N, nine large deficits can be observed at 590.3, 550, 475.6, 435, 230, 187.5, 115, 71.9 and 25 kyr BP. These can be assembled in groups corresponding to the four glacial periods of the Penck-Brückner scheme recognized by Köppen in the Milankovitch 65°N equivalent latitude.

These new results considering the reflective power of the polar caps in addition to the long-term variations of insolation were published in "Neue Ergebnisse" (Milankovitch, 1937-1938) and, according to Milankovitch are "absolutely sufficient to explain the full extent of even the greatest climatic events of the Quaternary and to clearly show their causes".

All these calculations show clearly that Milankovitch can be named the "father of paleoclimate modeling", certainly more specifically than the father of the astronomical theory in general (the first to propose the variations of the Earth's orbit as the causes of climate changes was Jens Esmark (1763-1839),

100 years before Milankovitch (Hestmark, 2017). Milankovitch was the first to demonstrate mathematically that the long-term variations of insolation are responsible for the climatic variations over the last hundreds of thousands of years. With his contemporary colleagues, Penck, Brückner, Köppen and Wegener (see also Thiede, 2018), to cite only a few among the most well-known, he could show that his mathematical climate fits well the geological reconstruction of climate available at that time.

## 7. Milankovitch, father of paleoclimate modeling

Milankovitch deserves this title of "father of paleoclimate modeling" because it is hard to find any scientist before him coming with so many papers stressing a mathematical climate and the calculation of the incoming solar radiation. He has written about 100 papers (in Serbo-Croatian/Cyrillic, German, or French, most of them having more than 20 pages) as a single author. His mathematical climate and his caloric incoming solar radiation are fully original, as well as his dating and interpretation of the paleoclimate data. To cite only his most well-known 1941 book, besides 8 chapters on fundamental celestial mechanics (150 pages) and 4 on polar wandering (61 pages), there are 3 chapters (80 pages) on Terrestrial insolation, 3 (70 pages) on connection between insolation and atmospheric temperature (his mathematical climate), and 5 (117 pages) on Ice ages, mechanisms and chronology. This is about 270 pages on what may be called paleoclimate modelling. Written in the first part of the 20th century, about 100 years ago, his work had a profound influence on the geologists and geophysicists of these early times (Penck, Brückner, Köppen and Wegener in particular), but continues to have one since the revival by Hays, Imbrie, Shackleton and Berger in the 1970s. The early part of the 20th century saw actually two exceptional geophysicists: Wegener (the father of continental drift, EOS-Transactions of AGU, vol 68 issue 19, p 516 by T.S. Ledley) and Milankovitch. Not only they worked together, but also both of them "had the imagination to bridge the confines of one discipline to make a major contribution to another."

Finally, it must be stressed that paleoclimate modelling is a subject more fundamentally related to astronomy (a forcing) than to $CO_2$ (a feedback). The title of father of paleoclimate modelling given to Milankovitch does not therefore make injustice to scientists like Joseph Fourier (1768-1830), John Tyndall (1820-1893), Svante Arrhenius (1859-1927), Guy Steward Callendar (1898-1964) and others. Their individual contributions to $CO_2$ must also be stressed. For example, the book on Théorie de la Chaleur (648 pages) by Fourier dates back 1822 and his paper on les "Températures du Globe terrestre" was published in 1824; the greenhouse effect by Tyndall dates back 1859, the year Arrhenius born and 37 years before the paper by Arrhenius. Such a paper on the history of $CO_2$ and the Ice Ages is certainly welcome to underline this other important aspect of paleoclimate (Bard, 2004) as well as a book on the full history of paleoclimates (Krüger, 2013).

**Code availability:** The insolation data used in Figure 1 can be calculated by using the code at https://www.elic.ucl.ac.be/modx/index.php?id=83.

**Competing interests**: The author declares that there is no conflict of interest.

**Special issue statement:** This manuscript is submitted to the special issue "A century of Milankovic's theory of climate changes: achievements and challenges (NPG/CP inter-journal SI)".

**Acknowledgments**

Thanks to Prof Vladimir Jankovic from the Center for the History of Science, Technology and Medicine at University of Manchester for his thoughtful comments. Thanks to Prof James Hays from the Lamont Doherty Earth Observatory, Columbia university, to Prof Martin Claussen from the Max-Planck Institute for Meteorology Hamburg and another anonymous reviewer for their helpful comments and proof-reading suggestions. Thanks also to Dr. Qiuzhen Yin from Université catholique de Louvain for reading the paper and for providing Figure 1.

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

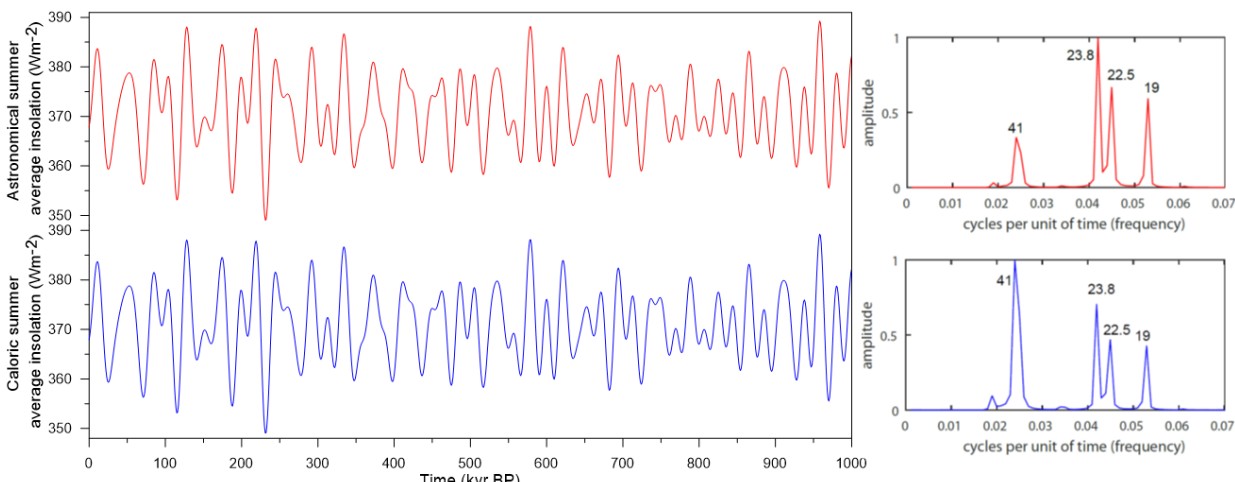

Figure 1: Average insolation of the half-year astronomical (red) and caloric (blue) summer seasons and their spectra (based on Berger and Loutre, 1991 and Berger et al., 2010). The major periodicities in kyr are indicated.