# Peer review of "Milankovitch, the father of paleoclimate modeling"

_Climate of the Past, 2021_

## Referee Comment (RC1)

Comments on the Manuscript:

Milankovitch, the father of paleoclimate modelling

By André Berger

"Contrary to what might be thought, Milankovitch has not calculated himself the long-term variations of the astronomical elements. He used them extensively for calculating the "secular march" of incoming solar radiation." These two sentences are, perhaps, the most important message of this manuscript. André Berger's clearly and concisely shows that Milutin Milankovitch is not the father of the astronomical theory of climate, but he is a paleoclimate modeler. He has advanced our understanding of Quaternary climate variations by two important and original contributions: the definition and use of caloric seasons and the concept of the "mathematical climate". This well written manuscript is a 'must read' for a special issue honoring the Milankovitch theory of Quaternary climate change.

Before publication, I would like to suggest considering one point, which might only be important for historical reasons. It does not touch the main message of the manuscript. On line 160 ff., it is stated that Milankovitch used the coloric seasons for the first time in his contribution (Mathematische Klimalehre) to the 1930 Köppen-Wegener Handbook. When going through the book "Die Klimate der geologischen Vorzeit" by Köppen and Wegener in 1924, I already find the concept of caloric seasons described by Milankovitch himself. Actually, Köppen and Wegener invited Milankovitch to contribute a piece of text in which Milankovitch himself cites his paper on "Kalorische Jahreszeiten und deren Anwendung im paläoklimalen Problem" (Ber. D. Königl. Serb. Adad. Bd. 1923). The text is very close to, albeit much shorter than, the text in Milankovitch's "Kanon der Erdbestrahlung" (1941). The defining figures (Figure 34 in "Die Klimate …" and Figure 42 in "Kanon …") are pretty much the same.

Regarding style, I would like to suggest rewriting the Abstract. When having read the Abstract, I frankly expected a rather technical paper. But likely, the manuscript is an eye opener for many paleoclimate modelers. Perhaps the two sentences I cited above, or similar sentences, could appear in the Abstract to make sure that everybody will read the paper. And it is a great pleasure to read it.

**Minor comments:**

Line 71: Not only the numerical values of the astronomical parameters were reproduced in Köppen and Wegener's "Die Klimate …", but also the insolation during the caloric summer and winter. The famous fold-up figure in "Die Klimate …" are already the caloric values, if I understood it correctly.

Line 142: Actually, Milankovitch himself wrote in his contribution to Köppen and Wegener's "Die Klimate …" : "Daß die folgende Tabelle sich auf die sommerliche Bestrahlung bezieht, geschieht auf Wunsch der Verfasser vorliegenden Werkes" (That the following table refers to summer irradiation is at the request of the authors of this work.) Hence, Köppen and Wegener claimed already in 1924 the summer half-year as decisive factor in glaciation – and they explicitly cited this as Penck's and Brückner's idea.

**Typos etc.:**

Title: paleoclimate modeling or palaeoclimate modelling? American or British English?

Line 38: Miskovitch or Miskovic? (In his "Kanon …", Milankovitch cited Miskovic as Michkovitch).

Line 48: $10^{-4}$, not 10-4

Line 62: Milankovitch's own words…  taken from which paper or book? (Actually, you can find them in both, "Die Klimate …" and the "Kanon …".)

Line 80 ff.  In the symbol $\Pi\gamma$ , the $\gamma$ should appear as subscript, if Milankovitch's "Kanon …" notation is followed, i.e., $\Pi_{\gamma}$ . Likewise, $\Pi_{\gamma}^{0}$ rather than $\Pi\gamma0$

Line 105, 107: which paper / book of Milankovitch's are these quotes taken from?

Line 120: interested in (not 'by') obliquity.

Line 144: Brückner instead of Bruckner.

Line 164: 'Canon' or 'Kanon', both quotes appear in the paper. I would harmonize it.

Line 187: Klimalehre (not Klimalehere), Stefan law (not Sefan law)

Line 188: IR? I guess it's infrared.

Line 205: Which "Köppen table" is meant?

Line 208 / 210: which paper / book is this quote taken from?

Line 239: amounted to 1920 caloric

Line 249: Neue Ergebnisse (not Neue Ergebnissen)

Line 294: Veränderungen der Bahnen der großen Planeten

Line 303: Klimate (not Kliamte)

Line 329: Erforschung der ….

Line 320: Klimalehre

Line 333: Erdbestrahlung

---

## Author Comment (AC1)

**Reply to reviewer 1**

Many thanks to Martin Claussen for his kind and useful comments. Please find here below my reply in black.

On line 160 ff., it is stated that Milankovitch used the caloric seasons for the first time in his contribution (Mathematische Klimalehre) to the 1930 Köppen-Wegener Handbook. When going through the book "Die Klimate der geologischen Vorzeit" by Köppen and Wegener in 1924, I already find the concept of caloric seasons described by Milankovitch himself. Actually, Köppen and Wegener invited Milankovitch to contribute a piece of text in which Milankovitch himself cites his paper on "Kalorische Jahreszeiten und deren Anwendung im paläoklimalen Problem" (Ber. D. Königl. Serb. Adad. Bd. 1923). The text is very close to, albeit much shorter than, the text in Milankovitch's "Kanon der Erdbestrahlung" (1941). The defining figures (Figure 34 in "Die Klimate …" and Figure 42 in "Kanon …") are pretty much the same.

I have cited the 1930 edition of Köppen-Wegener because I thought it was easier to find for the general reader. I have however added the reference 1924 because this edition was recently reproduced and translated into English by Thiede et al., Borntraeger, Stuttgart, 2015.

Köppen, W; and Wegener, A.: Die Klimate der geologischen Vorzeit. Borntraeger Berlin, 1924. Reproduction of the original German edition and complete English translation (The Climates of the Geological past) by Thiede J., Lochte K., Dummermuth ., Translated by Oelkers B., Borntraeger Stuttgart, 2015

I knew the original 1923 paper. It is actually a communication dating from 1922 (vol CVII) and the paper is written in Cyrillic. The original title and its translations in German, English and French are:

Kalorična godišnja doba I njihov primena u paleoklimaskom problemu; Kalorische Jahreszeiten und deren Anwendung im paläoklimalen Problem; Caloric seasons and their application in paleoclimate problem; Les saisons caloriques et leurs applications au problème paléoclimatique. In Separat iz., Glas Sprske kraljevske akademje, 1923, vol CIX, p. 1-30

Regarding style, I would like to suggest rewriting the Abstract. When having read the Abstract, I frankly expected a rather technical paper. But likely, the manuscript is an eye opener for many paleoclimate modelers. Perhaps the two sentences I cited above, or similar sentences, could appear in the Abstract to make sure that everybody will read the paper. And it is a great pleasure to read it.

I follow the suggestion of Martin Claussen and re-write the abstract:

**Abstract.** The history of the long-term variations of the astronomical elements used in paleoclimate research shows that, contrary to what might be thought, Milutin Milankovitch is not the father of the astronomical theory but he is definitely of paleoclimate modelling. He has not calculated himself these long-term variations but used them extensively for calculating the "secular march" of incoming solar radiation. He has advanced our understanding of Quaternary climate variations by two important and original contributions fully described in his Canon of Insolation. These are the definition and use of caloric seasons and the concept of the

"mathematical climate. How his mathematical model allowed him to give the caloric summer and winter insolation a climatological meaning is illustrated.

**Minor comments:**

Line 71: Not only the numerical values of the astronomical parameters were reproduced in Köppen and Wegener's "Die Klimate …", but also the insolation during the caloric summer and winter. The famous fold-up figure in "Die Klimate …" are already the caloric values, if I understood it correctly.

You are right

Line 142: Actually, Milankovitch himself wrote in his contribution to Köppen and Wegener's "Die Klimate …" : "Daß die folgende Tabelle sich auf die sommerliche Bestrahlung bezieht, geschieht auf Wunsch der Verfasser vorliegenden Werkes" (That the following table refers to summer irradiation is at the request of the authors of this work.) Hence, Köppen and Wegener claimed already in 1924 the summer half-year as decisive factor in glaciation – and they explicitly cited this as Penck's and Brückner's idea.

Yes I agree that Köppen, Wegener, Penck, Brückner and Milankovitch spread the Murphy's idea of a cool summer

**Typos etc.:**

Title: paleoclimate modeling or palaeoclimate modelling? American or British English?

I chose modelling

Line 38: Miskovitch or Miskovic? (In his "Kanon …", Milankovitch cited Miskovic as Michkovitch).

I use Michkovitch in this version of this article. In Milankovitch we find indeed *Miškovitch*. My writing was using by mistake 's' instead of '$\breve{s}$'and this last character is very difficult to reproduce.

Line 48: 10-4, not 10-4
OK

Line 62: Milankovitch's own words… taken from which paper or book? (Actually, you can find them in both, "Die Klimate …" and the "Kanon …".)
It is in his main books and contributions. I have added the references 1920 and 1941

Line 80 ff. In the symbol , the should appear as subscript, if Milankovitch's "Kanon …" notation is followed, i.e., . Likewise, rather than □□ □ □ □ 0□ □ 0□□
Done. It was actually well written in my manuscript

Line 105, 107: which paper / book of Milankovitch's are these quotes taken from?
MM 1941 Pp 264, 269, 270

Line 120: interested in (not 'by') obliquity.
Done

Line 144: Brückner instead of Bruckner.
Done

Line 164: 'Canon' or 'Kanon', both quotes appear in the paper. I would harmonize it.
I use Canon except in Kanon der Erdbestrahlung

Line 187: Klimalehre (not Klimalehere), Stefan law (not Sefan law)
Done

Line 188: IR? I guess it's infrared.
Yes I have written explicitely.

Line 205: Which "Köppen table" is meant?
It is the table of the altitude of the snow limit for different latitudes, see Figure 53 p 435 and table xxvii p 437 of MM 1941

Line 208 / 210: which paper / book is this quote taken from?
MM 1941 p 471

Line 239: amounted to 1920 caloric
OK it is canonic units

Line 249: Neue Ergebnisse (not Neue Ergebnissen)
Done

Line 294: Veränderungen der Bahnen der großen Planeten
Done

Line 303: Klimate (not Kliamte)
Done

Line 329: Erforschung der ….
Done

Line 320: Klimalehre
Done

Line 333: Erdbestrahlung

done

---

## Author Comment (AC2)

**Reply to reviewer 2**

Many thanks for the comments of Reviewer 2. Please find here below my reply in black.

palaeoclimatology and a corrective to misreadings of his climatological investigations.

I understand your concern about climate models, GCMs in particular. I agree with you that "These points could be considered in a more theoretical paper at some other time, and I do not think that this would be a place to resolve the questions." I hope that somebody will once write a nice history of modeling the global warming using in particular and mainly GCMs.

About Milankovitch, the reason why I think he deserves this title of "father of paleoclimate modeling" is because I have not found any scientist before him coming with so many papers stressing a mathematical climate and incoming solar radiation. He has written about 100 papers (most of them have more than 20 pages in Serbo-Croatian/Cyrillic, German, or French) as a single author which is totally different from the GCMs papers where there are so many authors that it becomes difficult to see their original individual contributions. His mathematical climate and his caloric incoming solar radiation are fully original, as well as his dating and interpretation of the paleoclimate data. To cite only his most well known 1941 book, besides 8 chapters on fundamental celestial mechanics (150 pages) and 4 on polar wandering (61 pages), there are 3 chapters (80 pages) on Terrestrial insolation, 3 (70 pages) on connection between insolation and atmospheric temperature (his mathematical climate), 5 (117 pages) on Ice ages, mechanisms and chronology. This is about 270 pages on what I call paleoclimate modelling; moreover, it was written in the first part of the 20$^{th}$ century, about 100 years ago. His work had a profound influence on the geologists and geophysicists of these early times (Penck, Brückner, Köppen and Wegener in particular) and continue to have one since the revival by Hays, Imbrie, Shackleton and berger. The early part of the 20$^{th}$century saw two exceptional geophysicists: Wegener (the father of continental drift, EOS-Transactions of AGU, vol 68 issue 19, p 516 by T.S. Ledley) and Milankovitch. Not only they worked together, but also both of them "had the imagination to bridge the confines of one discipline to make a major contribution to another."

Finally, I would like to stress that I suggest to call Milankovitch the "Father of PALEOLIMATE modeling". This a subject which is more related to astronomy (a forcing) than to $CO_2$ (a feedback). I therefore do not think that I do injustice to scientists like Arrhenius and others for whom I have a great respect. If we analyze the history of the $CO_2$ problem, like I did for the astronomical theories, we have at least: Fourier, Callendar, Tyndall and Arrhenius to discuss. I have been invited to participate to the symposia in honor of Arrhenius and of Tyndall and it is indeed not easy to decide who did the most important first contribution about $CO_2$. (the book on Théorie de la Chaleur (648 pages) by Fourier dates back 1822 and his paper on les "Températures du Globe terrestre" was published in 1824. The greenhouse effect by Tyndall dates back 1859, the year Arrhenius born and 37 years before the paper by Arrhenius). I wish a paper on the history of $CO_2$ will appear soon.